# Exploring the awareness, attitudes, and actions (AAA) of UK adults at high risk of severe illness from COVID-19

Stuart W. Flint[1,2], Adrian Brown[3,4]*, George Sanders[5], Abd A. Tahrani[6,7,8]

**1** School of Psychology, University of Leeds, Leeds, United Kingdom, **2** Scaled Insights, Nexus, University of Leeds, Leeds, United Kingdom, **3** Centre for Obesity Research, University College London, London, United Kingdom, **4** National Institute of Health Research, University College London, London, United Kingdom, **5** School of Sport, Leeds Beckett University, Leeds, United Kingdom, **6** Institute of Metabolism and Systems Research, University of Birmingham, Birmingham, United Kingdom, **7** Department of Diabetes and Endocrinology, University Hospitals Birmingham NHS Foundation Trust, Birmingham, United Kingdom, **8** Centre for Endocrinology, Diabetes and Metabolism, Birmingham Health Partners, Birmingham, United Kingdom

* a.c.brown@ucl.ac.uk

**Data Availability Statement:** The datasets used in the current study cannot be shared publicly because they contain potentially sensitive and identifiable patient information.Data are available

## Abstract

### Background

People at high risk of severe illness from COVID-19 have experienced greater restrictions during the pandemic, yet there is a paucity of research exploring their lived experience.

### Objectives

This study explored the impact of COVID-19 on people identified as at high risk of severe illness by UK Government, and in particular, the impact of the first lockdown on access to healthcare, medications and use of technological platforms.

### Methods

1038 UK adults who identified as at high risk of severe illness from COVID-19 in line with UK Government guidance or self-identified with acute or other chronic health conditions, completed the Awareness, Attitudes and Actions survey which explored the impact of COVID-19 on access to healthcare, management of long-term health condition, mental health, and health behaviours.

### Results

Most participants reported feelings of vulnerability, anxiety and isolation, noticed that other people changed their behaviour towards them including a feeling of being stigmatised by people not categorised as high risk. Participants described the largely negative impact that the COVID-19 lockdown had on to health-related behaviours and access to healthcare, which had resulted in large declines in mental health and wellbeing. Participants also indicated disappointment at the UK Governments response and handling of the COVID-19 lockdown.

on request from the University of School of Psychology Research ethic committee University of Leeds for researchers who meet the criteria for access to confidential data (psyc-ethicssubmissions@leeds.ac.uk).

**Funding:** This study was partly supported by an investigator-initiated grant from Ethicon, Johnson & Johnson to SWF; the funder had no role in study design, data analysis, data interpretation, or writing of the report (grant number: 202004001).

**Competing interests:** The authors have declared that no competing interests exist.

## Implications

This study provides novel evidence of the lived experience of the first COVID-19 lockdown for people identified as at high risk of severe illness. In the context of behavioural health interventions, the ubiquity of digital technologies and their adoption into day-to-day life translates into greater potential reach than traditional interventions, and consequently, greater potential for positive public health impact. Findings should be considered by policymakers and healthcare professionals to support people now and as we transition through the recovery phase with a particular emphasis on supporting mental health and changes to the management of long-term health conditions.

## Introduction

On 23 March 2020, the UK Government enacted measures that were included in the Coronavirus Act 2020 and recommended that everyone, except in certain essential circumstances, must stay in their homes [1]. Consequently, many of the elements enabling and supporting face-to-face access and healthcare became impossible to deliver due to the pandemic lockdown. In order to continue with everyday life, those identified as high risk had to self-isolate, shield and often adapt to solely digital or telephone solutions.

In addition to the population restrictions, it was recommended that people who were identified as at high risk of severe illness from COVID-19 follow greater restrictions compared to people without a high risk status; high risk status was based on an existing chronic health condition such as coronary heart disease, chronic kidney disease and diabetes, aged 70 years or above and if a person was pregnant [2]. With research demonstrating that people who were identified as at high risk were disproportionately impacted, and as countries across the world reduce restrictions such as national or regional lockdowns, there is a need to understand and appropriately respond to the experiences and changes that have occurred as a result. Gaining this knowledge should support changes that will need to be made to provide support for people who have been disproportionately impacted, and where relevant, to support the development and design of future care systems which have evolved during the pandemic such as the increased use of technology and changes to the way that people have accessed medications and healthcare.

Empirical research has demonstrated that despite concerns about infection of COVID-19, there is a lack of knowledge, use of risk mitigating behaviours or change to people's typical routines including amongst high risk groups [3, 4]. Furthermore, data published last year from our group identified that access to healthcare, health-related behaviours and psychological wellbeing of people at higher risk of severe illness from COVID-19 were negatively impacted [4]. This previous data demonstrates a quantified negative impact that people in high risk groups experienced during the first COVID-19 lockdown, however, there continues to be a dearth of qualitative research that explores the lived experience and provides in-depth accounts of the impact of the first COVID-19 lockdown.

The current study presents time-sensitive findings about the lived experience of people in high risk groups during the first COVID_19 lockdown in the UK, and in doing so, offers insights that can support the development or modification of existing care pathways, provision and policy in the UK.

## Methods

### Design

The Awareness, Attitudes and Action (AAA) survey was developed to explore the impact, including the lived experiences, of the COVID-19 pandemic and associated restrictions such as the national lockdown, on people identified as at high risk or self-identifying as at high-risk due to an acute or chronic condition not listed by UK Government; for further details of the AAA survey, please see Flint et al. 2020 [4]. The online survey was completed by 1026 UK adults aged 18 years or above who were recruited using pragmatic sampling [5] through a variety of advertisements disseminated by health charities, public and patient organisations, and on social media; for a breakdown of participant demographics, please see Flint et al. [4]. All participants provided informed consent before taking part in the study.

The AAA survey has 7 sections using both open-ended and closed questions. In our previous publication [4], we presented the findings of the closed questions which were analysed using statistical modelling and Behavioural Artificial Intelligence, and are published separately to the current study's findings [4].

The current study presents the analysis of the 17 qualitative, open-ended questions from the AAA survey. An example question was: *"Describe how being identified as being at a higher risk of severe illness from coronavirus (COVID-19) by the UK Government, has made you feel?"*; see S1 File for an overview of the open-ended survey questions. Consequently, questions demonstrated aspects of face validity, as they were transparent and relevant to the priority population [6]. Objectivity was maintained by the principal investigator as the resultant qualitative data aligned to the methodological framework for combining quantitative and qualitative survey methods [7], which was fit to serve as evidence for satisfying the research question [8]. Given the aim and objectives of this study, the Evidence Integration Triangle [9] was adopted as the overarching theoretical framework. Through the prompt identification of barriers and facilitators of reasons exploring AAA of UK adults at high risk of severe illness from COVID-19, the framework allows for the exploration of the three main evidence-based components of program/policy, implementation processes and measures of progress. Adoption of such an integrated framework allows for more consistent mapping, evaluation and incorporation of successful methods and strategies for modifying behavioural determinants [10].

We defined impact as changes that occurred as a consequence of restrictions to reduce COVID-19 transmission e.g., shielding, to different aspects of everyday life, including actions and attitudes, healthcare delivery, mental health and well-being, health-related lifestyle behaviours and social interaction.

Institutional ethical approval was received from the School of Psychology Research Ethics Committee at University of Leeds (REC number PSYC-28). All data were anonymised, and survey responses coded throughout to ensure confidentiality.

### Data coding and analysis

The pen profile approach presents findings from content analysis via a diagram of composite key emerging themes [11]. In summary, deductive content analysis was initially adopted to categorise survey data into four *a priori* pen profiles including i) awareness, ii) attitudes and concerns, iii) actions, and iv) changes. Inductive analysis then allowed emergent themes to be retrospectively applied into one of the four *a priori* pen profiles where relevant, with n representing individual 'mentions' per participant; multiple 'mentions' by the same participant were only counted once. Data were then organised schematically to assist with interpretation of the themes [12]. Verbatim quotations were subsequently used to expand the pen profiles,

provide context, and verify participant responses. Quotations were labelled by participant number (Pn). Methodological rigour was demonstrated through a process of triangular consensus between members of the research team. This offered transparency, credibility and trustworthiness of the results, as the data were critically reviewed using a reverse tracking process from the pen profiles back to the verbatim transcripts, providing alternative interpretations of the data. All investigators were in agreement with the initial interpretation of results made by the principal investigator.

## Results

The pen profiles and associated emergent themes are presented in Figs 1–4, respectively. Table 1 presents the emergent themes and frequency of mentions across the high risk groupings.

### Awareness

Fig 1 outlines participants' awareness and shows that the majority felt they had received sufficient information about the restrictions placed on high risk groups during the first lockdown.

> "*I had four letters informing me I was high risk and should shield, three similar letters from hospital consultants and one other from my GP.*"

There was a mixed response by participants when asked whether they agreed with their high risk group categorisation. Whilst many (64.3%) participants agreed with their high risk status, over a third (35.7%) of participants either disagreed or were unsure. Data suggested that one of the primary reasons for either agreeing or disagreeing was the available evidence to support high risk categorisation. This was further evidenced by participant perceptions about the

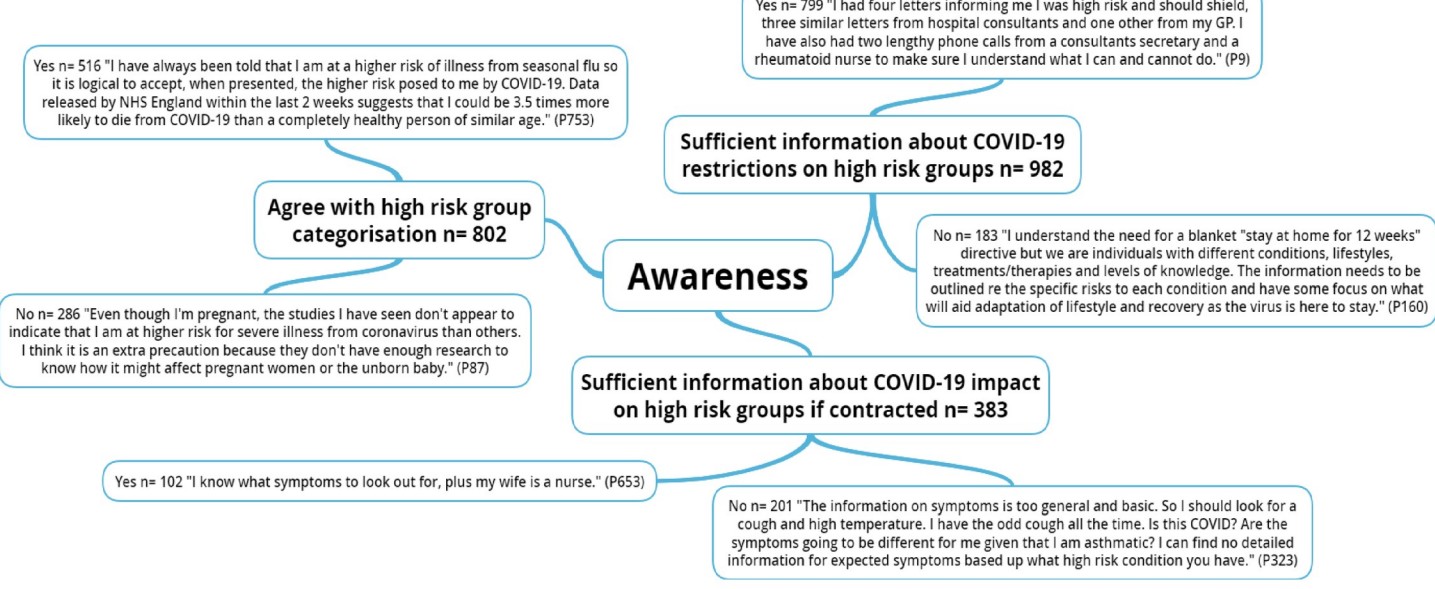

*n= Individual mentions per person (multiple mentions not included); Pn= Participant number

**Fig 1. Exploring the awareness of UK adults at high risk of severe illness during the COVID-19 lockdown.**

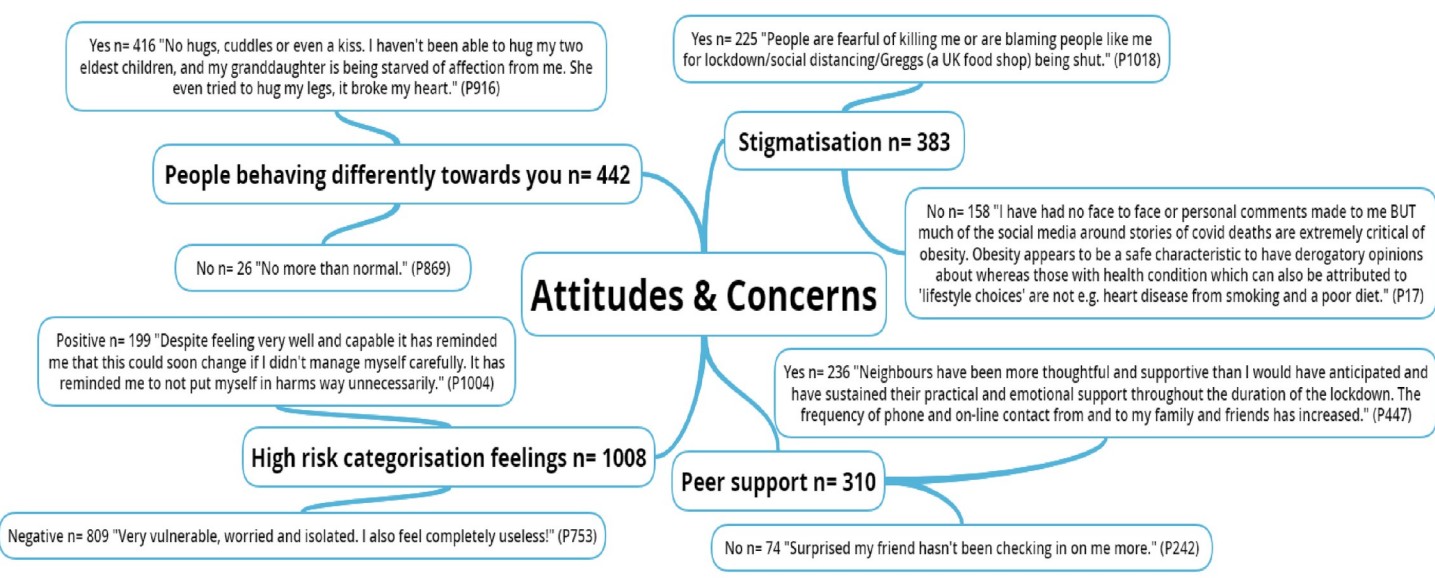

*n= Individual mentions per person (multiple mentions not included); Pn= Participant number.

**Fig 2. Exploring the attitudes and concerns of UK adults at high risk of severe illness during the COVID-19 lockdown.**

amount of information they had received about both restrictions for high risk groups and the health impact of contracting COVID-19.

> *"I have always been told that I am at a higher risk of illness from seasonal flu so it is logical to accept, when presented, the higher risk posed to me by COVID-19."*

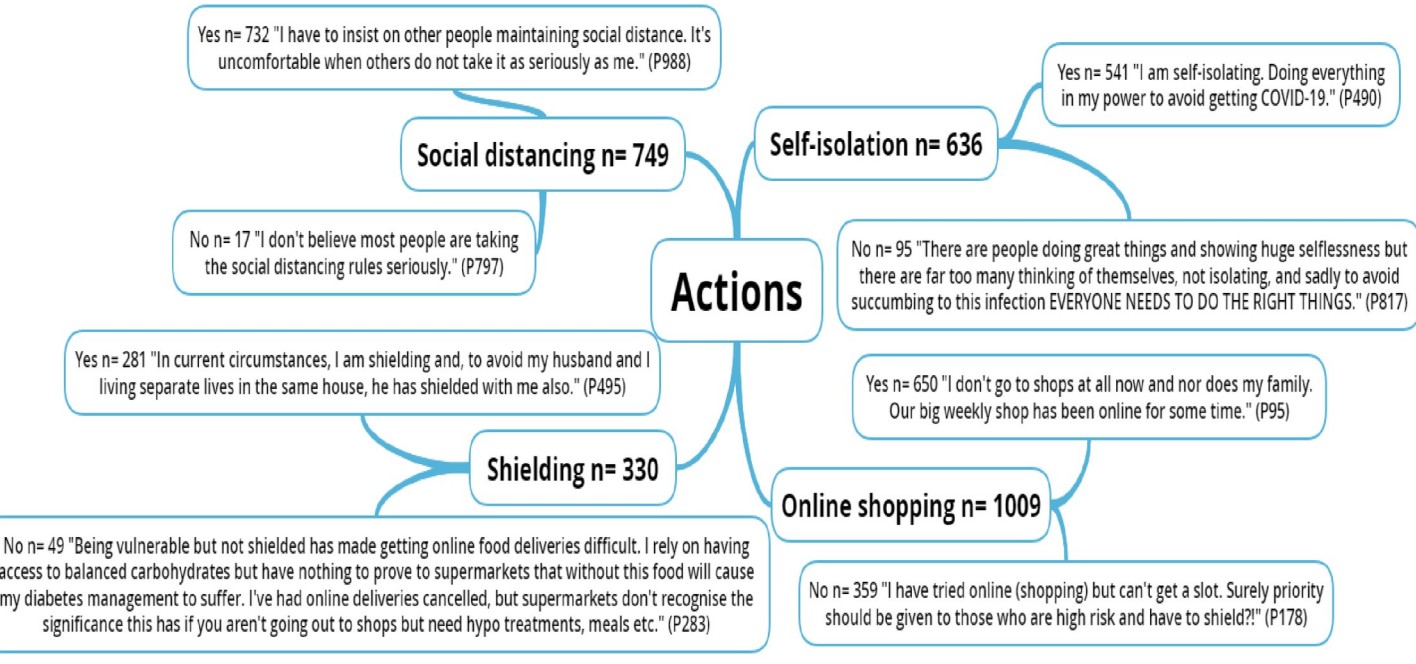

*n= Individual mentions per person (multiple mentions not included); Pn= Participant number.

**Fig 3. Exploring the actions of UK adults at high risk of severe illness during the COVID-19 lockdown.**

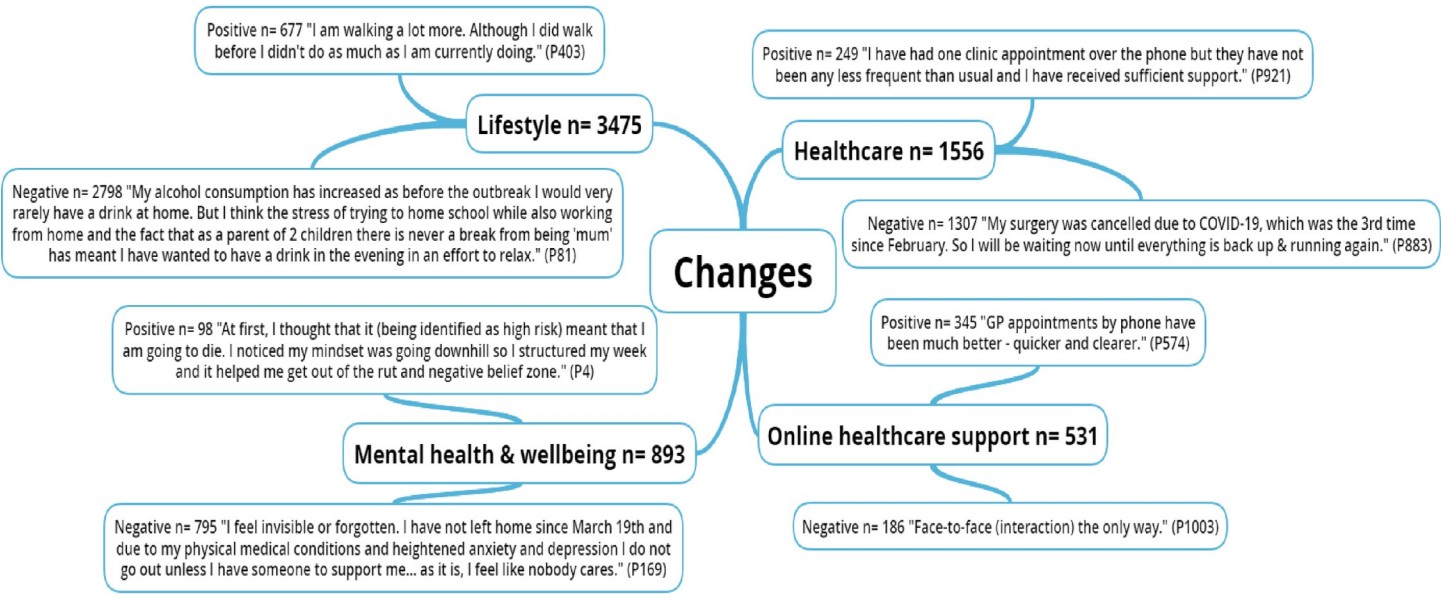

*n= Individual mentions per person (multiple mentions not included); Pn= Participant number.

**Fig 4. Exploring the changes of UK adults at high risk of severe illness during the COVID-19 lockdown.**

*"The information on symptoms (from COVID-19) are too general and basic"*

*"I think I should be on the shielding list because of my BMI and asthma, but the surgery are going by the government list which doesn't link medical conditions or include people with high BMI. I am very worried I won't survive if I get the virus as the asthma nurse told me my weight is affecting my ability to breathe. I am scared I will die"*

## Attitudes and concerns

Fig 2 outlines participants' attitudes and concerns towards contracting COVID-19. A large majority (94.1%) of respondents noticed people behaving differently towards them, and 58.7% of respondents reported feeling stigmatised. However, the majority (76.1%) received sufficient peer support during the first COVID-19 lockdown.

*"Kept away physically and stopped communicating by phone or any other medium. There's more blame, aggression and dismissiveness."*

*"The frequency of phone and on-line contact from and to my family has increased."*

Furthermore, participants noted largely negative (80.3%) emotions towards being categorised as high risk with feelings which centred on vulnerability, anxiety, and isolation, which in some instances led to further feelings of stigmatisation. Participants not only commented that they felt vulnerable, but that the high risk status was having an impact on their mental health.

*"I live alone and feel very isolated yet nobody has explained anything to me. My family live a long distance away and I am feeling very lonely and sad at not knowing when I may be given some vague timeframe when I can see my daughter and Granddaughters. I think my mental health is suffering very badly."*

**Table 1. Emergent themes: Frequency of mentions across high risk sub-groups.**

| Theme | Frequency of sub-group mentions (Yes:No), unless otherwise stated. | | | | | | | | | | | |
|---|---|---|---|---|---|---|---|---|---|---|---|---|
| | Diabetes (N = 538) | BMI ≥ 40 kg/m² (N = 142) | Chronic Respiratory Disease (N = 179) | Chronic Heart Disease (N = 132) | Chronic Kidney Disease (N = 147) | Chronic Liver Disease (N = 49) | Chronic Neurological Disease (N = 35) | Spleen Problems (N = 16) | Weakened Immune System (N = 159) | Aged > 70 years (N = 178) | Pregnant (N = 21) | Other Risk Factors (N = 303) |
| **Awareness** | | | | | | | | | | | | |
| Agree with high risk categorisation | 146:25 | 35:9 | 46:14 | 35:11 | 40:18 | 18:12 | 17:9 | 5:4 | 24:89 | 53:45 | 10:11 | 87:39 |
| Sufficient restriction information | 151:44 | 73:21 | 71:11 | 64:13 | 53:12 | 14:6 | 11:8 | 3:0 | 19:2 | 131:34 | 5:7 | 204:25 |
| Sufficient impact information | 102:90 | 41:12 | 72:17 | 52:21 | 40:19 | 14:6 | 11:4 | 7:0 | 86:22 | 44:57 | 8:5 | 39:33 |
| **Attitudes & Concerns** | | | | | | | | | | | | |
| Behave differently towards you | 69:6 | 74:9 | 35:0 | 27:0 | 17:0 | 15:5 | 0:3 | 8:0 | 44:0 | 86:3 | 14:0 | 27:0 |
| Stigmatisation | 65:42 | 25:19 | 21:17 | 4:12 | 0:11 | 0:0 | 0:0 | 0:8 | 24:0 | 37:13 | 5:0 | 44:36 |
| Peer Support | 18:4 | 23:6 | 8:3 | 21:1 | 22:0 | 9:1 | 14:0 | 0:1 | 31:4 | 55:19 | 11:7 | 24:28 |
| High risk categorisation feelings (+ve:-ve) | 20:262 | 25:105 | 18:14 | 13:49 | 15:23 | 7:14 | 0:19 | 0:11 | 19:77 | 57:104 | 5:13 | 20:118 |
| **Actions** | | | | | | | | | | | | |
| Social distancing | 214:3 | 56:0 | 71:0 | 66:0 | 53:0 | 14:0 | 7:0 | 0:0 | 71:0 | 98:10 | 9:0 | 73:4 |
| Self-isolation | 87:18 | 62:24 | 68:0 | 54:0 | 34:0 | 15:0 | 21:0 | 4:0 | 52:0 | 102:0 | 7:6 | 35:47 |
| Online shopping | 124:37 | 65:22 | 89:28 | 36:19 | 32:25 | 28:21 | 18:17 | 16:0 | 52:68 | 96:53 | 9:5 | 85:64 |
| Shielding | 22:7 | 25:0 | 14:4 | 28:3 | 17:0 | 7:12 | 6:0 | 11:0 | 85:12 | 33:1 | 2:0 | 31:10 |
| **Changes** | | | | | | | | | | | | |
| Lifestyle (+ve:-ve) | 304:884 | 77:397 | 18:141 | 35:177 | 24:53 | 31:95 | 15:72 | 8:26 | 33:103 | 27:341 | 7:22 | 98:487 |
| Healthcare (+ve:-ve) | 49:373 | 23:188 | 31:107 | 0:104 | 22:62 | 22:59 | 15:15 | 0:28 | 12:141 | 42:121 | 0:17 | 33:92 |
| Online healthcare support (+ve:-ve) | 68:11 | 34:13 | 29:34 | 37:8 | 14:0 | 18:12 | 7:14 | 0:7 | 45:20 | 32:39 | 0:0 | 61:28 |
| Mental health & wellbeing (+ve:-ve) | 12:181 | 9:104 | 3:86 | 4:88 | 17:61 | 5:33 | 3:20 | 3:7 | 13:81 | 22:63 | 4:4 | 3:67 |

Note: +ve, positive -ve, negative.

## Actions

Fig 3 shows that, on the whole, participants adhered to Government recommendations regarding social distancing (97.7%), self-isolation (85.1%) and shielding (85.1%). Although there was a sense of resentment to those not following the recommendations and not taking them seriously, while others felt that everyone should be playing their part in reducing risk.

> *"I am following all the precautionary guidelines so as to not infect anybody and also to keep me and my family safe."*

Participants also noted utilising online shopping methods during the first COVID lockdown to support adherence to such guidelines. However, issues related to online shopping were noted in 35.6% of participant comments, where participants were unable to use online shopping due to a lack of availability, with a feeling that priority should be given to those at highest risk and with chronic diseases.

> *"Our big weekly shop has been online for some time."*

> *"I have tried online (shopping) but can't get a slot."*

> *"Determined to try not to get it, but feel those of us in the high risk group have been abandoned by the Government. I have read reports of people in that group still being forced to work by their employer, in high risk roles. Many of us can't get online shopping slots, even those who have shopped for years online. I've seen first-hand stories of disabled people who have paid for carers withdrawn, due to illness, when they can't physically look after themselves."*

## Changes to healthcare and health-related lifestyle behaviours

Fig 4 outlines largely positive (65%) views regarding the switch from face-to-face to online or telephone healthcare support. Contrastingly, negative views were expressed with regards to the changes that the first COVID-19 lockdown caused to health-related lifestyle behaviours (80.5%) (across shopping, diet, alcohol consumption, physical activity, & sleep) and healthcare access (84%) (across appointments, medication, elective surgery, communication platform, & clinician care), as well as mental health and wellbeing (89%).

> *"Less exercise leading to increased weight. A sense of anxiety about becoming ill. A feeling that if I catch it [COVID-19] I will die as I am of no further use to society. A big fear of not surviving until a vaccine is readily available. A worry that the vaccine may not work fully as it is being produced in a big hurry."*

> *"I am very concerned about the delay in being scanned [for cancer]. I have worries about the return of the cancer as I wasn't able to detect anything myself when it was found, therefore, it could be back and I would have no knowledge of it."*

These delays resulted in many participants independently changing the management of their long-term health condition and in some instances, without notifying their healthcare provider.

> *"I've stopped taking Adalimumab (Amgevita 40mg fortnightly). Not discussed with medics/ specialist nurses as feel they will express alarm and castigate me., whereas it's a relatively*

*informed decision and ultimately it's about my body and my life. Typing this is making me think I probably should have discussed it with health team"*

Finally, what was also evident throughout participant's responses was disappointment at the UK Governments response and handling of the COVID-19 lockdown. Many participants reported that high concern about the UK Government's response. For instance,

*"The UK government was slow to act initially even though it had clear examples across Europe of how the pandemic would spread. I found its advice vague and optional. It has put the immediate economy ahead of lives. Cummings has destroyed the credibility of lockdown. I am angry at how this has all been handled."*

*"Don't feel the government acted quickly or strongly enough around lock down. The situation with airports is shocking."*

*"Afraid, insecure, very anxious, stressed, verging on depression (i am prone to it and use HRT/ CBT to balance it). I assume if I get COVID-19 then I will likely die. the fact we have an utterly incompetent government who are making things worse on purpose whilst they trial their 'herd immunity plan' with no cure doesn't help me feel that it the situation will get better any time soon. Given that I also have a heart condition, the stress from this isn't helping."*

Within this theme, were reoccurring negative perceptions about the UK Government's handling of the Dominic Cummings incident (Former Chief Advisor to the UK Prime Minister; [13]). Comments included:

*"I feel that the lack of immediate response by the government, the mumbling, bumbling style of our prime minister and the disgraceful behaviour of Dominic Cummings have considerably increased my anxiety levels."*

*". . . the government has decided to distract us all from the Dominic Cummings fiasco by loosening lockdown by too much too soon"* and that *"steps to ease lockdown are being rushed to deflect attention from the Dominic Cummings story and not based on sound science. If only we lived in NZ."*

*"I felt very comfortable that things were under control until last weekend when the government let me down and indicated that the rules don't need to be followed (thanks to Mr Cummings escapades). I now feel a lot less secure".*

*"Dominic Cummings might have known the rules for exceptional circumstances but these were not publicised to the general public beyond STAY AT HOME. Thank you."*

## Discussion

This study is the first to provide a comprehensive qualitative exploration of the lived experiences of people identified as at high risk of severe illness by the UK Government. The study provides insights into their awareness, attitudes and actions relating to healthcare, health-related behaviours and mental health during the first COVID-19 lockdown in the UK, and as such, provide rich information about the unintended consequences of the COVID-19 restrictions. The results from this qualitative analysis reinforce and extend the insights from our previously published study [4]. Specifically, this study provides in-depth data about the lived experience and perceptions of people identified as at high risk of severe illness from COVID-

19 during the first lockdown. In doing so, contributes further to our understanding of the impact of COVID-19 on mental health and wellbeing, health-related behaviours and reduced access to healthcare. The reduced access to healthcare reported by participants in this study has been substantiated by official UK Government statistics, showing that during the first wave of the pandemic, hospital admissions, accident and emergency attendances and the number of GP consultations were all down on previous years [14]. As reported by some participants in the current study, the reduction in people with worsening health conditions were attributed to a desire to not contribute to the pressure on the healthcare service as well as concerns about becoming infected with COVID-19 [14]. The perceptions reported of wanting to avoid further pressurising the NHS reflect the consistent media messaging that the healthcare service was struggling to cope with the demand, and in some cases, may have led to missed or delayed treatment to manage acute and chronic health conditions.

This study demonstrated that people identified as high risk of severe illness elicited feelings of vulnerability, anxiety, and isolation, which in some led to feelings of stigmatisation and blame for the COVID-19 restrictions. As such service providers and the Government should consider the potential implications of future messaging relating to COVID-19 and indeed messaging relating to other public health topics, as well as how these unintended consequences can be minimised and avoided. Most participants in this study noticed other people behaving differently towards them, and alarmingly, over half of respondents also reported feeling stigmatised. As we transition during the pandemic, it is imperative that people who are at high risk are supported to re-participate in society given the greater restrictions they have experienced during the pandemic and can do so without bias or discrimination. For instance, a pertinent area to consider is the return to work given the importance of employment for health and wellbeing [15].

Furthermore, results showed that participants adhered to UK Government recommendations regarding social distancing, self-isolation and shielding. However, there was a sense of resentment to those not following the recommendations and not taking them seriously, while others felt that everyone should be playing their part in reducing risk. Previous research has reported that resistance to change is not uncommon where choices and aspects of freewill are required to take a sudden change of approach, reflecting the rapid changes that resulted from the decisions undertaken by the UK Government during the COVID-19 lockdown [16]. To overcome such resistance to change, it is important that social support networks are available [16]. The current study highlighted mostly positive comments regarding support and participants felt they had the necessary contact from family and friends which in turn, helped them engender a culture where they felt confident to make decisions and take actions. Further efforts could also be made by the UK Government to adopt a bottom-up approach to information dissemination, whereby a greater effort is made to voice the opinions of those at ground level [17].

Findings of particular concern was that participants reported reduced access to healthcare provision (across appointments, medication, elective surgery, communication platform, & clinician care) which resulted in them independently changing the management of their long-term health condition, potentially placing them at even greater risk. This, in turn, had a detrimental impact on mental health and wellbeing, supporting recent research examining the impact on people at high risk of COVID-19 infection [18–20]. This highlights that services need to consider how they can support their patients during periods of reduced access, to ensure they feel supported and remain safe. This may involve self-management support strategies in primary care to upskill patients to manage their own conditions, these strategies have been shown to improve clinical indicators, self-efficacy and patient knowledge [21]. COVID-19 has challenged the way staff have been taught to deliver care, as roles and responsibilities

have been redefined, new tools and processes implemented, and cross-professional/sectoral collaboration formalised [17]. This can often lead to resistance, resignation or disregard. Existing literature suggests that both trust-based (relationship dynamics) and control-based (organisational dynamics) governance mechanisms play a crucial role in partnership development [22]. There is widespread agreement that a bottom-up approach is required, whereby the purpose and benefits of the change should not only be understood and embraced by individuals and communities at ground level, but also coproduced with them [23]. Given the speed of transition needed during the first lockdown this was not possible. However, lessons learned from this study and others need adopting beyond the current pandemic to ensure competency for future lockdown situations among as significant a proportion of the population as possible.

The COVID-19 pandemic exacerbates the importance of a hidden form of social inequality, namely digital inequalities [24]. Digital services have great potential to improve population health and the efficiency and reach of basic needs and healthcare [25]. Mobile apps, text messages, wearable and ambient sensors, social media, and interactive websites can improve health by supporting behaviours involved in disease prevention, self-management of long-term conditions and delivery of evidence-based practice [25]. Such was reflected in the current study as the adoption of digital services appeared to be received positively by the majority of participants and expands on our understanding of how digital services have been received. Availability, ease of access and quality of support were the major benefits noted and were ubiquities across all high risk groups. The qualitative data did not seem to explain why women did not like digital services as much [4], and therefore this needs further exploration and whether this might impact on service uptake that have a higher prevalence of female patients. Although many participants were aware and often accepted that a transition to digital solutions was compulsory, concurrent with previous research [26], there was a preference to return to 'normal' face-to-face interactions as soon as possible. Hence, hybrid solutions that offer a blend of face-to-face and digital treatment tailored, as best as possible, to individual needs may be the most effective solution in the future. A focus on equity and ethics is warranted to ensure digital health truly increases access to impactful digital services. Indeed, it's important to ensure that the provision of digital services does not worsen inequalities [27]. Engagement approaches that help individuals of all backgrounds be aware and understand the benefits of digital services including online campaigns, social media exposure, and radio advertising are warranted.

This change in the way healthcare is provided is likely to continue in the future and is in line with the NHS long-term plan for increased digital healthcare provision [28], with older participants being particularly keen on digital healthcare continuing [4]. This is in line with previous research showing that older participants ($\geq$55 years of age) demonstrate considerable interest in learning how to use the internet for accessing services [29]. However, service providers' ambitions to engage with older adults online appear more limited as a result of entrenched stereotypes of older non-users, a lack of internal digital skills, as well as organisational and funding constraints [29]. The current study's findings emphasise the importance of balancing the views of older adults and service providers in the design of online engagement strategies. These insights are critical for improving online service delivery affected by an increasing withdrawal of traditional services. Further research is warranted in exploring the best methods to deliver training and support for digital services to enable as many individuals as possible access. Harnessing the surge in interest, enthusiasm, and acceptance of digital offers during lockdown situations has immediately been recognised as an opportunity for service providers [30]. Consequently, complementary qualitative evaluations are crucial to fully understand and interpret user experiences, as well as developing and evaluating user engagement and overall effectiveness.

The perceptions of people identified as high risk relating to the UK Government response, including the Dominic Cummings incident, is particularly pertinent given the guidance that specified greater restrictions on their daily life. Clearly, this incident led to a national outcry where some people believed that Cummings' actions—that are akin to other lockdown guidance breaches and led to other officials leaving their posts—has both impacted people's emotional response and raised questions of Government handling of the incident [31]. At a time when national action and strong leadership was required [32], breaches by the very proponents and perceived leaders of lockdown legislation may undermine its continued implementation and negatively affect people most impacted.

A strength of the study was the comprehensive assessment of qualitative survey data theoretically underpinned by the methodological framework for combining quantitative and qualitative survey methods [7]. The triangulation of data is a further strength which enhanced understanding of qualitative survey data and subsequently, overall awareness, attitudes and actions. Furthermore, this study addresses the limitation identified in previous research about only using polling data to understand public opinions regarding lockdown policies [33]. Study limitations are also noted and discussed in previous publications [4]. A pragmatic sub-sample of people identified as at high risk of severe illness from COVID-19 were recruited via convenience sampling methods and hence results cannot be considered generalisable. The subjective nature of the data is also a limitation, as is the presence of self-selection bias which resulted from the pragmatic sampling methods adopted.

## Conclusion

The findings of this study provide novel insights about the lived experience of UK adults at high risk of severe illness during the first COVID-19 lockdown. In the context of digital service offers such as accessing basic needs and healthcare, the ubiquity of digital technologies and their adoption into day-to-day life translates into greater potential reach, and consequently greater potential for positive impact. However, the potential public health impact of these digital service offers can only be realised to the extent of their availability, accessibility, and efficacy. These rich findings sheds new light on the lived experiences of COVID-19 for UK adults identified as at high risk, which should be considered by policymakers, healthcare and charities and other organisations supporting people in these high risk groups both now, and through the recovery phase of the pandemic.

## Supporting information

**S1 File. Awareness, Attitudes and Actions (AAA) survey.**
(DOCX)

**S1 Dataset.**
(XLS)

## Acknowledgments

We would like to acknowledge and thank the following organisations for disseminating the survey.

APPG Obesity
APPG Medical Research
Anticoagulation UK
Association for Medical Research Charities
Asthma UK

Blackpool, Fylde & Wyre haemochromatosis Support Group
British Dietetics Association
British Liver Trust
British Lung Foundation
Diabetes UK
Elton John AIDS Foundation
European Association for the Study of Obesity
Hepatitis C Trust
Kidney Care UK
JDRF
Leeds Academic Health Partnership
LIVErNORTH
Melanoma UK
MS Society
Nexus Leeds
Norfolk & Norwich Liver Group
Obesity UK
Parkinson's Society
PBC Foundation
Public Health Scotland
Research for the Future
Salford Sickle Cell Society
Terrance Higgins Trust
The Somerville Foundation
Yorkshire Cancer Community

## Author Contributions

**Conceptualization:** Stuart W. Flint.

**Data curation:** Stuart W. Flint.

**Formal analysis:** Stuart W. Flint, George Sanders.

**Funding acquisition:** Stuart W. Flint.

**Investigation:** Stuart W. Flint, Adrian Brown, Abd A. Tahrani.

**Methodology:** Stuart W. Flint, Adrian Brown, Abd A. Tahrani.

**Project administration:** Stuart W. Flint, Adrian Brown, Abd A. Tahrani.

**Software:** George Sanders.

**Writing – original draft:** Stuart W. Flint, George Sanders.

**Writing – review & editing:** Stuart W. Flint, Adrian Brown, George Sanders, Abd A. Tahrani.

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
