## [Decision Letter · Decision Letter 0]

10 Aug 2021

PONE-D-21-15375

Exploring the awareness, attitudes, and actions (AAA) of UK adults at high risk of severe illness from COVID-19

PLOS ONE

Dear Dr. Brown,

Thank you for submitting your manuscript to PLOS ONE. After careful consideration, we feel that it has merit but does not fully meet PLOS ONE’s publication criteria as it currently stands. Therefore, we invite you to submit a revised version of the manuscript that addresses the points raised during the review process.

We look forward to receiving your revised manuscript.

Kind regards,

Christophe Leroyer

Academic Editor

PLOS ONE

“This study was partly supported by an investigator-initiated grant from Ethicon, J&J; the funder had no role in study design, data analysis, data interpretation, or writing of the report.”

 “This study was partly supported by an investigator-initiated grant from Ethicon, J&J to SWF; the funder had no role in study design, data analysis, data interpretation, or writing of the report.”

6. Please note that in order to use the direct billing option the corresponding author must be affiliated with the chosen institute. Please either amend your manuscript to change the affiliation or corresponding author, or email us at plosone@plos.org with a request to remove this option.

7. Thank you for submitting the above manuscript to PLOS ONE. During our internal evaluation of the manuscript, we found significant text overlap between your submission and the following previously published works, some of which you are an author.

https://bmjopen.bmj.com/content/10/12/e045309.info

Please revise the manuscript to rephrase the duplicated text, cite your sources, and provide details as to how the current manuscript advances on previous work. Please note that further consideration is dependent on the submission of a manuscript that addresses these concerns about the overlap in text with published work.

Additional Editor Comments (if provided):

Reviewers' comments:

Reviewer's Responses to Questions

**Comments to the Author**

1. Is the manuscript technically sound, and do the data support the conclusions?

Reviewer #1: Yes

Reviewer #2: Yes

2. Has the statistical analysis been performed appropriately and rigorously? 

Reviewer #1: Yes

Reviewer #2: Yes

3. Have the authors made all data underlying the findings in their manuscript fully available?

Reviewer #1: Yes

Reviewer #2: Yes

4. Is the manuscript presented in an intelligible fashion and written in standard English?

Reviewer #1: Yes

Reviewer #2: Yes

5. Review Comments to the Author

Reviewer #1: My answer is generally very favorable, but one point needs clarification. The author repeatedly associates "mental health" and "well-being". So there seems to be a closeness between the two terms as well as a difference. Could he be more specific?

Reviewer #2: Dear Authors,

Thanks for submitting your article to PlosOne. Thank you for giving me the opportunity to read your very interesting work.

The subject of this article is impact of COVID-19 lockdown in a high risk of severe illness UK patients. It is a relevant subject. Authors used a specific questionnaire described in a previous article (BMJ Open 2020). This is a qualitative study on answers from this survey. Methods are relevant. Few data on this field have been published. Tables and results are well described. The article is well structured.

Aims of the study are clearly written in introduction and answers well described in discussion and conclusion

English language is good.

Minor revisions:

In Introduction p 4 l76-78: introduction or methods. I think it is relevant to have a specific sentence on your definition of the term impact in methods.

Table 1: in the part attitudes and concerns and also self-isolation be careful on tabulation

Defined +ve and – ve in legend

P 13 l 261 put a ref (maybe not scientific) on Dominic Cumming’s incident or explain it in some words for non UK readers…

Discussion:

Shortened the discussion

Some ref could be added as:

Psychological distress during the COVID-19 pandemic in France: a national assessment of at-risk populations.

Chaix B, Delamon G, Guillemassé A, Brouard B, Bibault JE. Gen Psychiatr. 2020 Nov 26;33(6):e100349. doi: 10.1136/gpsych-2020-100349. eCollection 2020.

PMID: 34192239 Free PMC article.

The limitations of polling data in understanding public support for COVID-19 lockdown policies.

Foad CMG, Whitmarsh L, Hanel PHP, Haddock G. R Soc Open Sci. 2021 Jul 7;8(7):210678. doi: 10.1098/rsos.210678. eCollection 2021 Jul.

PMID: 34258021 Free PMC article.

1. Understanding the psychological impact of the COVID-19 pandemic and containment measures: An empirical model of stress.

Wissmath B, Mast FW, Kraus F, Weibel D.

PLoS One. 2021 Jul 29;16(7):e0254883. doi: 10.1371/journal.pone.0254883. eCollection 2021.

PMID: 34324498 Free PMC article.

2. Impact of COVID-19-like symptoms on occurrence of anxiety/depression during lockdown among the French general population.

Mary-Krause M, Herranz Bustamante JJ, Héron M, Andersen AJ, El Aarbaoui T, Melchior M.

PLoS One. 2021 Jul 26;16(7):e0255158. doi: 10.1371/journal.pone.0255158. eCollection 2021.

PMID: 34310661 Free PMC article.

6. PLOS authors have the option to publish the peer review history of their article (what does this mean?). If published, this will include your full peer review and any attached files.

Reviewer #1: **Yes: **Anne-Hélène Le Cornec Ubertini, Senior Lecturer in Information and Communication Sciences

Reviewer #2: No

---

## [Author Response · Author response to Decision Letter 0]

5 Oct 2021

Thank you very much for the editors and reviewers useful comments which we have addressed them below. Please find our responses in bold. 

Thank you for your comment we have reviewed the PLOS ONE style requirements and amended accordingly. 

Thank you for the comment this has been added to the Funding information section as requested as follows: 

“This study was partly supported by an investigator-initiated grant from Ethicon, Johnson & Johnson to SWF; the funder had no role in study design, data analysis, data interpretation, or writing of the report (grant number: 202004001)”

“This study was partly supported by an investigator-initiated grant from Ethicon, J&J; the funder had no role in study design, data analysis, data interpretation, or writing of the report.”

 “This study was partly supported by an investigator-initiated grant from Ethicon, J&J to SWF; the funder had no role in study design, data analysis, data interpretation, or writing of the report.”

This has been removed from the main text and has been added to the reply letter as follows:

“This study was partly supported by an investigator-initiated grant from Ethicon, Johnson & Johnson to SWF; the funder had no role in study design, data analysis, data interpretation, or writing of the report (grant number: 202004001)”

Thank you for your comment. We have added the study minimal data set to the resubmission find attached. The qualitative data within the study offers in some instances personal and sensitive experiences during COVID-19 e.g., about personal clinical experiences with clinician and about medical conditions, which could result in people being identified. But can be made available for researchers who meet the criteria for access to confidential data on request to the School of Psychology Research ethic committee University of Leeds

We have changed the data sharing statement as recommended to the following: 

"The datasets used in the current study cannot be shared publicly because they contain potentially sensitive and identifiable patient information. Data are available on request from the University of School of Psychology Research ethic committee University of Leeds for researchers who meet the criteria for access to confidential data (psyc-ethicssubmissions@leeds.ac.uk)"

As above, we have added the study minimal data set to the resubmission find attached.

6. Please note that in order to use the direct billing option the corresponding author must be affiliated with the chosen institute. Please either amend your manuscript to change the affiliation or corresponding author, or email us at plosone@plos.org with a request to remove this option.

Dr Adrian Brown is affiliated with the University College London and should be the corresponding author. There was no place within the revision section of journal platform to confirm this. But have indicated on the manuscript that Dr Brown is the corresponding author and submitted the manuscript. 

Thank you for your comment. We have checked the referencing and added further references in response to changes to the reviewers comments. 

7. Thank you for submitting the above manuscript to PLOS ONE. During our internal evaluation of the manuscript, we found significant text overlap between your submission and the following previously published works, some of which you are an author. 

https://bmjopen.bmj.com/content/10/12/e045309.info

Please revise the manuscript to rephrase the duplicated text, cite your sources, and provide details as to how the current manuscript advances on previous work. Please note that further consideration is dependent on the submission of a manuscript that addresses these concerns about the overlap in text with published work.

Thank you for your feedback and for noting that these sections needed to be addressed in particular in the introduction and methods. We have significantly revised the text to avoid any overlap with the previous publication, please see update manuscript. 

1. Please include captions for your Supporting Information files at the end of your manuscript, and update any in-text citations to match accordingly. Please see our Supporting Information guidelines for more information: https://eur01.safelinks.protection.outlook.com/?url=http%3A%2F%2Fjournals.plos.org%2Fplosone%2Fs%2Fsupporting-information&data=04%7C01%7Ca.c.brown%40ucl.ac.uk%7C35cc163fee454a82452408d981f1e66a%7C1faf88fea9984c5b93c9210a11d9a5c2%7C0%7C0%7C637683698214095605%7CUnknown%7CTWFpbGZsb3d8eyJWIjoiMC4wLjAwMDAiLCJQIjoiV2luMzIiLCJBTiI6Ik1haWwiLCJXVCI6Mn0%3D%7C1000&sdata=%2Bo0eMgfOiavaUpH47CermovLzFfv5Y5m5VUoayEKL3I%3D&reserved=0.

Thank you for the comment. We have now included a caption at the end of the manuscript for the supplementary materials called: S1 Supplementary Information: Awareness, Attitudes and Actions (AAA) survey. This is identified within the text where needed. 

Reviewer 1: 

We thank this Reviewer for their helpful comments regarding our manuscript and we have addressed each point below. 

My answer is generally very favorable, but one point needs clarification. The author repeatedly associates "mental health" and "well-being". So there seems to be a closeness between the two terms as well as a difference. Could he be more specific?

Thank you for your useful and insightful comment. Mental health and wellbeing are interrelated terms. Mental health is defined as “a positive concept related to the social and emotional wellbeing of individuals and communities”. Wellbeing can be wider to include physical, social as well as psychological wellbeing. Therefore we have used the term “mental health and wellbeing” to be inclusive of both concepts.

Reviewer 2: 

We thank this Reviewer for their helpful comments regarding our manuscript and we have addressed each point below. 

Dear Authors,

Thanks for submitting your article to PlosOne. Thank you for giving me the opportunity to read your very interesting work.

The subject of this article is impact of COVID-19 lockdown in a high risk of severe illness UK patients. It is a relevant subject. Authors used a specific questionnaire described in a previous article (BMJ Open 2020). This is a qualitative study on answers from this survey. Methods are relevant. Few data on this field have been published. Tables and results are well described. The article is well structured.

Aims of the study are clearly written in introduction and answers well described in discussion and conclusion

English language is good.

Minor revisions:

In Introduction p 4 l76-78: introduction or methods. I think it is relevant to have a specific sentence on your definition of the term impact in methods.

Thank you for your comment we have moved the sentence within the introduction to the methods and clarified how we are defining impact. 

Table 1: in the part attitudes and concerns and also self-isolation be careful on tabulation 

Thank you for the comment we have reviewed the table and the column and rows appear to be appropriately formatted. But we have add addition lines to help 

Defined +ve and – ve in legend

Thank you we have added in this to the note page line 

P 13 l 261 put a ref (maybe not scientific) on Dominic Cumming’s incident or explain it in some words for non UK readers…

Thank you for your comment we have clarified who Dominic Cumming was and also referenced a correspondence published in the Lancet on the incident.

Discussion:

Shortened the discussion 

Thank you for your comment. We have reviewed the discussion and shortened the discussion. Please see revised manuscript. 

Some ref could be added as: 

Thank you for your recommendation for adding references we have reviewed them and added two of the suggested recommendations that were appropriate. 

Psychological distress during the COVID-19 pandemic in France: a national assessment of at-risk populations. 

Chaix B, Delamon G, Guillemassé A, Brouard B, Bibault JE. Gen Psychiatr. 2020 Nov 26;33(6):e100349. doi: 10.1136/gpsych-2020-100349. eCollection 2020.

The limitations of polling data in understanding public support for COVID-19 lockdown policies. 

Foad CMG, Whitmarsh L, Hanel PHP, Haddock G. R Soc Open Sci. 2021 Jul 7;8(7):210678. doi: 10.1098/rsos.210678. eCollection 2021 Jul.

---

## [Editor Report · Decision Letter 1]

19 Oct 2021

Exploring the awareness, attitudes, and actions (AAA) of UK adults at high risk of severe illness from COVID-19

PONE-D-21-15375R1

Dear Dr. Brown,

We’re pleased to inform you that your manuscript has been judged scientifically suitable for publication and will be formally accepted for publication once it meets all outstanding technical requirements.

Kind regards,

Christophe Leroyer

Academic Editor

PLOS ONE

---

## [Editor Report · Acceptance letter]

26 Oct 2021

PONE-D-21-15375R1 

Exploring the awareness, attitudes, and actions (AAA) of UK adults at high risk of severe illness from COVID-19 

Dear Dr. Brown:

I'm pleased to inform you that your manuscript has been deemed suitable for publication in PLOS ONE. Congratulations! Your manuscript is now with our production department. 

Kind regards, 

on behalf of

Dr. Christophe Leroyer 

Academic Editor

PLOS ONE